# Validation of SuPerSense, a Sensorized Surface for the Evaluation of Posture Perception in Supine Position

**DOI:** 10.3390/s23010424

**Published:** 2022-12-30

**Authors:** Daniela De Bartolo, Ilaria D’amico, Marco Iosa, Fabio Aloise, Giovanni Morone, Franco Marinozzi, Fabiano Bini, Stefano Paolucci, Ennio Spadini

**Affiliations:** 1Scientific Institute for Research, Hospitalization and Health Care (IRCCS) Santa Lucia Foundation, Via Ardeatina 306, 00179 Rome, Italy; 2Department of Mechanical and Aerospace Engineering, Sapienza University of Rome, Via Eudossiana 18, 00184 Rome, Italy; 3Department of Psychology, Sapienza University of Rome, Via Dei Marsi 78, 00185 Rome, Italy; 4Alfameg s.r.l., Via Giacomo Favretto 9, 00147 Rome, Italy; 5Department of Life, Health and Environmental Sciences, University of L’Aquila, 67100 L’Aquila, Italy; 6Perceptive Lab s.r.l., Via Sebastiano Veniero 22, 00192 Rome, Italy

**Keywords:** body schema, spatial cognition, load cells, center of pressure, stabilometry

## Abstract

This study aimed to validate a sensorized version of a perceptive surface that may be used for the early assessment of misperception of body midline representation in subjects with right stroke, even when they are not yet able to stand in an upright posture. This device, called SuPerSense, allows testing of the load distribution of the body weight on the back in a supine position. The device was tested in 15 patients with stroke, 15 age-matched healthy subjects, and 15 young healthy adults, assessing three parameters analogous to those conventionally extracted by a baropodometric platform in a standing posture. Subjects were hence tested on SuPerSense in a supine position and on a baropodometric platform in an upright posture in two different conditions: with open eyes and with closed eyes. Significant correlations were found between the lengths of the center of pressure path with the two devices in the open-eyes condition (R = 0.44, *p* = 0.002). The parameters extracted by SuPerSense were significantly different among groups only when patients were divided into those with right versus left brain damage. This last result is conceivably related to the role of the right hemisphere of the brain in the analysis of spatial information.

## 1. Introduction

Stroke is a cerebrovascular disease caused by the total or partial lack of blood supply to the brain [1]. Depending on the extent of damage caused by the brain lesion, the longitudinal after-stroke course may vary significantly from one patient to another. Different outcomes are possible, from patients with a mild compromise of gait to those totally dependent on a wheelchair, usually affected by a hemiparesis of the body side contralateral to the damaged brain hemisphere [1,2].

Therefore, after a stroke, one of the obstacles to rehabilitation is functional asymmetry, which reduces the ability to walk and maintain an upright posture, which is also affected by poor balance. Evidence suggests that the upright posture in stroke patients is characterized by weight-bearing asymmetry, that is, the shifting of the load onto the non-paretic leg, thus causing greater postural oscillations. Previous studies have shown that a biased perception of body symmetry [3], motor weakness [4,5], asymmetric muscular tone, [6,7], and somatosensory deficits [7] may participate in the deficits related to postural instabilities and asymmetries [8].

In addition, spatial cognitive disorders may also be involved in this kind of asymmetry since the misperception of both egocentric and allocentric frames of reference could result in the distortion of the coordinates used to distribute loading on both legs while standing [9]. 

In this regard, an assessment of these asymmetries as well as the beginning of a rehabilitative intervention should occur as soon as possible. According to the scientific literature [9,10,11,12], medical devices and instruments that evaluate postural asymmetry are needed even before patients can return to their feet; in this way, timely intervention is possible, which is usually positively associated with good rehabilitation outcomes in patients with stroke [13]. 

In line with this objective, we developed a sensorized version of the SuPer, which is a Surface for Perceptive Rehabilitation [10,11,12]. The previous non-sensorized version has already successfully been used to assess and rehabilitate patients with low back pain [11] and patients with fibromyalgia [12] by recovering correct postural control starting from a supine posture. Furthermore, this tool has also been used in neurological populations, such as patients with Parkinson’s disease, to assess and treat postural balance [10]. SuPer is a therapeutic system based on the interaction between the patient’s back and a support surface that is comprised of small latex cones of various dimensions and elasticities. In the non-sensorized version, these cones were applied with their inferior bases on a rigid wood surface through elastic strips, and usually, over 100 cones were used for each session. Patients were asked to lay supine on the surface that was formed by the smoothed apex of these cones, creating reaction forces to the patient’s weight generated by the interaction with the cones. These forces produced high pressure on the small area of contact, resulting in many intensive perceptive stimuli that helped provide patients with augmented feedback on their supine posture [10,11,12].

In this paper, we propose the validation of the SuPerSense©, a smart sensorized version of the SuPer, to obtain important clinical information about possible asymmetries in patients with stroke. Considering that postural asymmetry is not only due to the weakness of the lower limb contralateral to the brain lesion but also a misperception of the body midline [3,9], we think that the SuPerSense could provide important information about the postural assessment of patients at different stages of disease, even in a supine position. 

Since no previous studies have addressed this aspect in the literature, we compared the evaluation in an orthostatic posture, which can be easily detected via a baropodometric pressure surface, to that in a supine posture recorded through the SuPerSense.

## 2. Materials and Methods

All measurements and experimental conditions were performed in full accordance with the Declaration of Helsinki standards for human research. The study was approved by the local independent ethics committee of the IRCCS Santa Lucia Foundation (Rome, Italy). Every subject provided written informed consent prior to participation. We recruited patients with stroke according to the following inclusion/exclusion criteria and two control groups of healthy subjects: one of young subjects to obtain physiological values and one of healthy elderly participants to allow an age-matched comparison with patients.

### 2.1. Participants

For this study, we enrolled three groups of participants: 15 healthy young participants (young group, YG), 15 patients with stroke (patient group, PG), and 15 healthy subjects age-matched with the PG as controls (elderly group, EG). 

The inclusion criteria for stroke participants were:-Age over 18 years;-Certified diagnosis of ischemic or hemorrhagic stroke by a neurological medical examination;-Ability to stand alone or with the help of a physiotherapist in open- and closed-eyes conditions (OE, CE) for at least one minute.

The exclusion criteria for patients with stroke were:-Presence of cognitive impairment or linguistic difficulties that made it difficult to understand and follow the instructions of the experimenter;-Presence of pathological comorbidities (such as neurological, psychiatric, orthopedic, and metabolic diseases) that may affect motor performance under examination.

For healthy subjects, the inclusion criterion was age between 20 and 30 years for YG and between 45 and 75 years for EG, and the exclusion criterion was the presence of cognitive or motor deficits.

### 2.2. Equipment

Supine posture was recorded through a smart surface (SuPerSense© System, Perceptive Lab s.r.l., Rome, Italy). It consists of a polyethylene platform on which 126 removable latex cones were arranged (Figure 1). The latex cones are positioned on surfaces through Velcro application. 

Three types of cones are included in this system: 18 are passive and equipped to close the circuit, allowing no introduction of noise; 42 cones are cap surfaces and inserted to provide the patients with a comfortable laying position. The remaining 66 are logic active cones (expandible up to 84, including the passive ones) on which there is a conditioning and a load cell responsible for detecting the weight exerted on them. Active cones are powered by a control unit connected by means of a flexible board present in the lower part of the sensorized surface. Every single cell transmits the value to the control unit, which collects and processes it and then sends it to the control software. 

The control unit connects the computer with the weight sensors through a Bluetooth connection or USB cable. 

Data are acquired through software implemented in the Java environment that allows the visualization and recording of the data coming from the sensors installed on the surface. This software runs on a PC, tablet, or smartphone, and thus the use of this device is versatile. 

Hence, the main difference between SuPerSense and the previous version of SuPer is the sensorized cones positioned on a load cells. Each load cell was characterized by a capacity of 10 kg, a rated output of 1.0 (s.d. 0.2) mV/V, and a repeatability of 0.1% F.S. The distance between two cells (and hence between the apices of cones) was equal to 5.5 cm. An additional difference is that the cones in the SuPerSense system have the same dimensions and the same elasticity. 

Upright posture was assessed with a baropodometric and stabilometric platform (FreeMed, Sensor Medica). This device is a plantar support evaluation system for static, dynamic, and stabilometric acquisitions that has been used in previous studies on neurologic populations [14,15]. 

### 2.3. Procedure

Each subject was assessed individually in a unique recording session. For each participant, personal, anthropometric, and clinical information was collected. Before undergoing the experimental procedure, participants wearing heavy clothing and patients with medical braces were asked to take off them to optimize the interaction between the skin and the surfaces for both supine and upright posture recordings. 

The latter was investigated by carrying out a Romberg test [15,16], during which each participant was asked to stand barefoot on the platform with their arms aligned with their hips while looking at an optical sight set in front of them about one meter away. We made two recordings—one with open eyes (OE) and one with closed eyes (CE)—each lasting 51.2 s at a sampling rate of 25 Hz according to a previous study [14]. 

For supine assessment, participants were asked to lie on the SuPerSense system with the midline of their trunk aligned with that of the device (Figure 1) so that their shoulders were positioned on the upper side of the surface and their pelvis was positioned on the lower one. We recorded two trials for the two experimental conditions (OE and CE) of 60 s, which were then cropped to match the duration of the recordings in the upright position, using a sampling rate of 1 Hz, according to the manufacturer’s recommendations. 

Between the two trials, the smart surface was calibrated in the absence of contact.

### 2.4. Data Collection and Parameters 

(CoP) data were collected using the FreeMed baropodometric platform (SensorMedica, Rome, Italy) with dimensions 120 cm × 50 cm and a sampling frequency of 400 Hz and Free Step software. This software also allows the processing of raw data into stabilometric parameters on which it is not necessary to carry out other pre-processing computations (such as the application of filters). According to the literature [15,17], the chosen parameters for this analysis were the length of the CoP path, which represents the global trajectory of the center of pressure on the platform, and the 95% confidence ellipse area, expressed as the percentage of load on one foot compared to the other.

Every parameter was computed as in previous studies [15,17] The global percentage load on each foot was used for the calculation of the symmetry index, which indicated the alignment of the weight distribution with respect to the median line in the standing position. Similarly, in a previous study on stroke patients [18], the ratio of the minimum value between the right mean load and left mean load divided by the maximum value was chosen as the index of symmetry.
(1)$$Symmetry\;Index=\frac{min\;\left(load_{\;left},\;load_{right}\right)\;}{max\;\left(load_{\;left},\;load_{right}\right)}*100$$

The data collected by the SuPerSense system were saved by the software associated with the device and organized in a .csv file of 127 columns, of which 126 represented the cones and one showed the instants of time recorded during the acquisition of the signal. Therefore, for each recording, the final file contained the data on the weight recorded by the load cells expressed in grams. This allowed the computation, in the supine position, of the timely position of the center of pressure and the analogous parameters evaluated by the baropodometric platform in upright standing posture: the length of the CoP path, the area of the ellipse containing 95% of the CoP positions, and the symmetry index. The data collected through the SuPerSense Surface were imported into Matlab for the calculation of stabilometric indices.

All the parameters taken into consideration were analyzed according to the two experimental conditions (OE and CE).

### 2.5. Statistical Analysis

The normal distribution of each calculated parameter was verified using the Shapiro–Wilk test. Not all parameters were found to be normally distributed (*p* < 0.05); therefore, non-parametric statistics were performed for the inferential analyses. The Kruskal–Wallis test was used to perform comparisons between groups with respect to the parameters described above: the length of the CoP path, the area of the ellipse, and the symmetry index. The analyses were performed considering the 3 main groups—the PG, YG, and EG—and the 4 groups obtained after dividing the PG into two subgroups: patients with damage in the right hemisphere of the brain and those with damage in the left one. Specific analyses between the PG and EG were conducted using the Mann–Whitney U test. Finally, to assess whether there was a correlation between the indices obtained with the baropodometric platform and the SuPerSense Surface, an overall analysis was carried out on the whole group with the Spearman correlation coefficient, choosing a level α of statistical significance equal to *p* < 0.05.

For all the analyses, the α level of statistical significance was set at *p* < 0.05. All analyses were performed using IBM SPSS Statistics for Windows software (Version 28.0. IBM Corp., Armonk, NY, USA).

## 3. Results

We enrolled 45 subjects (26 females, 58%), 15 for each group. Table 1 reports the demographical and anthropometric characteristics of the enrolled samples, no statistically significant differences were noted between PG and EG. Ten patients had an ischemic stroke (67%) and five had a hemorrhagic event (33%), nine of them in the right hemisphere of the brain (60%) and six in the left one (40%).

Figure 2 shows the mean values of the three analyzed parameters for the three groups in the open-eyes and closed-eyes conditions. The trends observed for the baropodometry, with higher values recorded for the PG with respect to the other two groups, which had values similar to each other, were also mainly observed for the corresponding parameters recorded with the SuPerSense, especially in the OE condition. In the CE condition, only the ellipse area maintained the same trend, whereas similar values were recorded among groups for the length of the CoP path and symmetry index.

No statistical differences were noted for the SuPerSense when all patients were analyzed as a single group, but when they were subdivided according to the damaged hemisphere, the CoP path length and the ellipse area in the open-eyes condition showed significant differences among groups.

Analyzing all subjects, significant correlations were found between some parameters recorded on the SuPerSense and the baropodometric parameters. In particular, the SuperSense length of the CoP path in the open-eyes condition was significantly correlated with the analogous parameter measured on the baropodometric platform (R = 0.44, *p* = 0.002). The CoP ellipse area measured by the SuperSense was significantly correlated with all the baropodometric parameters (R > 0.38, *p* < 0.011). The same parameter measured in the closed-eyes condition resulted was related to the length of the CoP path measured by the platform in the open-eyes (R = 0.40, *p* = 0.006) and closed-eyes (R = 0.34, *p* = 0.022) conditions and the ellipse area (R = 0.64, *p* < 0.001) in the open-eyes condition. The SuPerSense symmetry index did not show a significant correlation with the platform parameters. All results are reported below in Table 2.

## 4. Discussion

The aim of this study was to validate the SuPerSense, a device for the assessment of posture in a supine position. Our results showed that, as expected, the differences between patients and healthy subjects were reduced in the supine position with respect to the upright position. However, similar trends were observed, especially in the open-eyes condition, and statistically significant correlations between parameters assessed with baropodometry and the analogous ones assessed with SuPerSense were observed. The neuro-motor processes related to balance rely on afferent sensory input from both sides of the body and visual space, the motor output provided to lower limbs and trunk, and a high-level body representation related to the anthropometric body midline. The latter is sensitive to hemispheric asymmetry, which is usually altered in stroke as well as other neurological disorders [10]. The impairments in the integration of somatosensory inputs may increase axial kinesthetic impairments [19,20]. In stroke, as shown for patients with Parkinson’s disease, a misperception of the body midline could alter balance control in a particular fashion unrelated to the motor deficits [21], and for this reason, it could be possible to assess it in the supine position [10]. Conversely, the importance of correct midline perception for a correct representation of the patient’s body orientation in space was enlightened in previous studies by Karnath and colleagues on neglect syndrome [22,23]. 

When we divided our group of patients according to the affected hemisphere, we found even more interesting (and statistically significant) results. Patients with a stroke in the right hemisphere showed higher instability even in the supine position (higher CoP path length and ellipse area) when tested in the open-eyes condition. It is well known that damage in the right hemisphere of the brain can cause extrapersonal [24] and/or personal [25,26] unilateral spatial neglect. Although our patients were not affected by neglect, right brain damage can impair the spatial representations of the body midline that guide actions within the peripersonal and extrapersonal spaces, controlling the posture [27]. It is interesting that, in our study, the differences observed for patients with right brain damage were statistically significant only in the open-eyes condition. A possible explanation is related to an attentional shift in the midline perception when the visual system is involved in the analysis of body posture in the space versus the condition with closed eyes, in which most of the information is related to the tactile feedback provided by the SuPerSense on the back of the patient. 

However, body midline perception and related back motion and postural problems [12] are altered not only in neurological conditions, such as those that occur in subjects affected by PD and stroke, but also in all chronic back pain conditions [10,12]. 

Although often underestimated in rehabilitation, body midline perception has received more attention in cognitive sciences and general neurosciences. The concept of the midline of the body is important in human primates, as demonstrated by an fMRI study by Fabri et al. [28]. In this research, the authors demonstrated that the cutaneous regions adjacent to the trunk midline are represented bilaterally in the first somatic sensory cortex [28]. 

Cyte et al., in an interesting study on the influence of gravity on the orientation and localization of the body midline [29], highlighted that the correct perception of the body midline in terms of orientation and localization comes out from the integration of sensory information. Thus, the authors suggested that the vestibulo-ocular reflex (VOR), and in particular its horizontal component, contributed to the perceived localization of the body midline. By contrast, tactile information mainly influenced body midline orientation.

In addition, body midline perception can also be manipulated through external unilateral stimuli [29], opening the doors to an important rehabilitation opportunity. In fact, the rehabilitative effect of SuPer was based on increasing the back tactile feedback that led to an augmented perception and or reeducation of the body midline, as reported in previous studies [10,11,12]. However, further studies should investigate this difference to clarify the underlying neural and sensorial mechanisms.

A limitation of our study is related to the intrinsic differences between assessment in the supine and upright positions. However, the absence of a gold standard for the assessment of the supine position led us to test our patients in these two very different conditions. Another limitation is the reduced sample size, especially when two subgroups of patients were considered; further studies are needed with larger samples of patients to compare those with right brain damage with those with left brain damage. 

## 5. Conclusions

The SuPerSense is a sensorized version of the SuPer, a surface for perceptive rehabilitation of the back trunk that has already been used in previous rehabilitative studies [10,11,12]. The SuPerSense could provide objective parameters to assess the supine position of patients analogous to those provided by baropodometric platforms for the upright standing posture. 

This is the first study to have investigated supine posture, but so much more needs to be done. Despite the reduced sensitivity for the common parameters usually investigated in the upright posture, our study advanced the field of postural assessment regarding:

(1) The possibility of analyzing postural asymmetries for patients in the early stage after the stroke characterized by a certain postural instability, which usually impairs their independence in standing up on their own.

(2) The possibility of exploring postural asymmetries with respect to midline misperception.

(3) The ability to carry out early assessments of specific sensory and representational deficits, especially in patients with stroke in the right hemisphere of the brain. 

## 6. Patents

The SuPerSense was patented in Italy on 18 September 2020 with number n.102020000022033. 

## Figures and Tables

**Figure 1 sensors-23-00424-f001:**
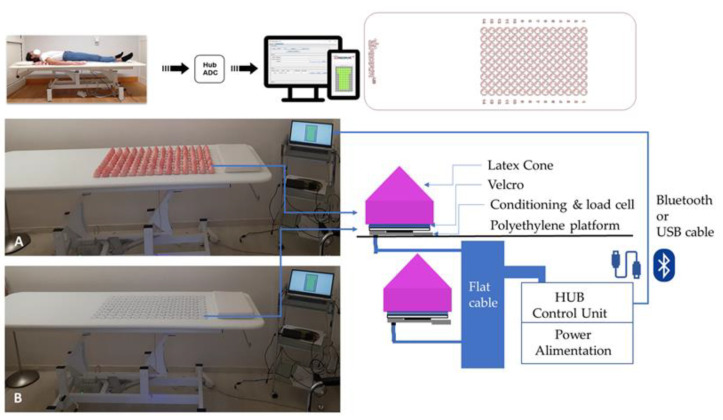
Experimental set-up of SuPerSense sensorized surface. As shown in the figure, in the top left, a subject is lying on the surface; the top right of the figure shows the model of the load cells. Above, there are two close pictures of the device with (**A**) and without (**B**) latex cones. Near the pictures is a schematic representation of how the device is made and works. The software included with the device allows recordings to be taken via both computer and tablet.

**Figure 2 sensors-23-00424-f002:**
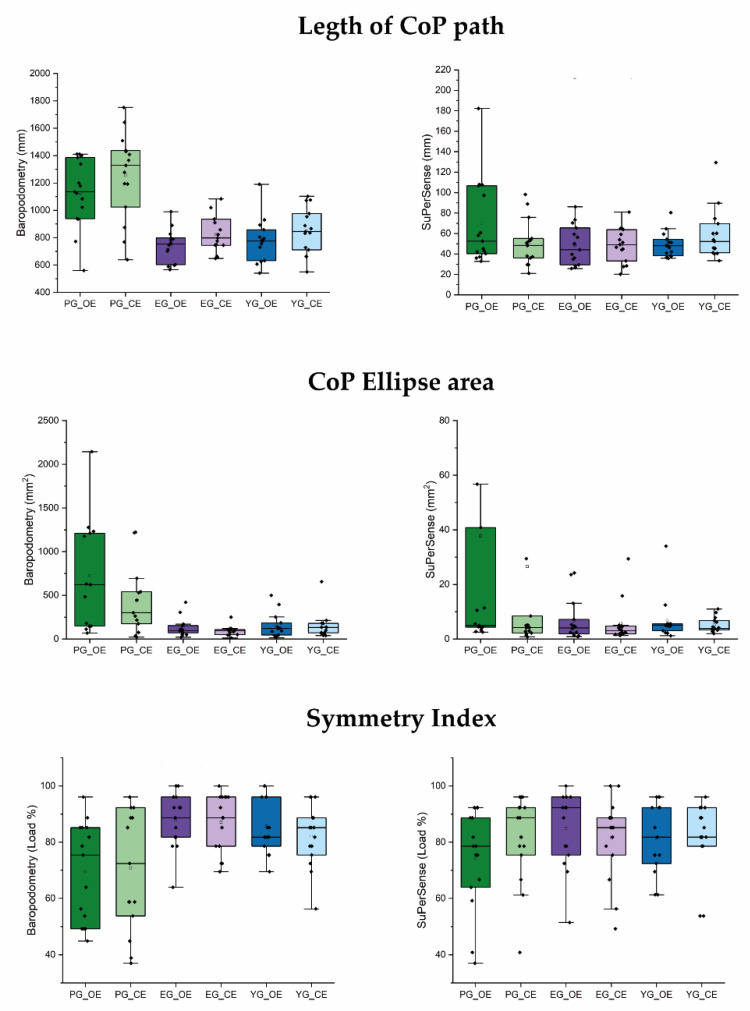
Box plot figure. Each left panel is the specific index measured with baropodometric force plate, and the right ones are those recorded with the SuPerSense Surface system. Black and small square stands for the mean, the line flat inside the box (which comprises the 25~75% of distribution) is the median. Vertical line between dashes is the delimitation for the range within the 1.5 IQR (intra-quarter range). Green represents the patient group (PG), violet represents the elderly group (EG), and blue represents the young group (YG). For each group, the darkest colors are associated with the open-eyes condition (OE), while light colors are associated with the closed-eyes condition (CE).

**Table 1 sensors-23-00424-t001:** Demographic and anthropometric characteristics of the young group (YG), elderly group (EG), and patient group (PG). Measures are summarized as the mean ± standard deviation values. The last two columns report the *p*-values for the comparison among groups (Kruskal–Wallis analysis, KW) and between the PG and EG (Mann–Whitney U test).

	YG (*n* = 15)	EG (*n* = 15)	PG (*n* = 15)	KW Analysis	PG vs. EG
**Age (years)**	24.7 ± 3.2	57.2 ± 9.1	60.3 ± 10.4	*p* < 0.001	0.267
**Stature (cm)**	169.0 ± 9.0	169.7 ± 7.4	173.9 ± 10.5	*p* = 0.268	0.174
**Body Weight (kg)**	60.9 ± 10.8	72.9 ± 19.5	75.2 ± 14.5	*p* = 0.021	0.436
**BMI (m^2^/kg)**	17.9 ± 2.5	21.4 ± 5.0	21.5 ± 3.2	*p* = 0.016	0.653

**Table 2 sensors-23-00424-t002:** Results of statistical analysis. Comparisons among the three groups are made using the Kruskal–Wallis test, while specific comparisons between stroke patients (PG) and control group (EY) are performed using the Mann–Whitney U test. Bold results stand for significant statistical difference.

Condition	Parameter	YG (n = 15)	EG (n = 15)	PG L (n = 6)	PG R (n = 9)	*p*-Value 3 Groups	*p*-Value 4 Groups
**Baropodometry**	**CoP Path length (mm)**	770 ± 162	741± 118	1098 ± 301	1145 ± 231	**<0.001**	**<0.001**
**Open Eyes**	**CoP Ellipse Area (mm^2^)**	211 ± 235	238 ± 295	523 ± 411	788 ± 706	**0.004**	**0.006**
	**Symmetry Index (%)**	86 ± 10	88 ± 10	73 ± 22	69 ± 18	**0.002**	**0.005**
**Baropodometry**	**CoP Path length (mm)**	851 ± 166	823 ± 132	1159 ± 355	1320 ± 288	**<0.001**	**<0.001**
**Closed Eyes**	**CoP Ellipse Area (mm^2^)**	156 ± 150	177 ± 334	318 ± 250	458 ± 461	0.063	0.102
	**Symmetry Index (%)**	82 ± 11	87 ± 10	78 ± 20	66 ± 22	**0.017**	**0.018**
**SuperSense**	**CoP Path length (mm)**	49 ± 12	48 ± 19	53 ± 24	81 ± 49	0.061	**0.023**
**Open Eyes**	**CoP Ellipse Area (mm^2^)**	7 ± 8	7 ± 7	6 ± 4	59 ± 86	0.071	**0.010**
	**Symmetry Index (%)**	82 ± 12	85 ± 13	68 ± 18	78 ± 17	0.137	0.115
**SuperSense**	**CoP Path length (mm)**	59 ± 25	49 ± 17	54 ± 21	46 ± 24	0.410	0.527
**Closed Eyes**	**CoP Ellipse Area (mm^2^)**	5 ± 3	6 ± 7	3 ± 1	42 ± 78	0.190	0.054
	**Symmetry Index (%)**	82 ± 13	81 ± 15	80 ± 23	83 ± 9	0.997	0.981

## Data Availability

Raw data are available upon specific requests to the corresponding author.

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
