# Peer review of "Validation of SuPerSense, a Sensorized Surface for the Evaluation of Posture Perception in Supine Position"

_sensors, 2022, doi:10.3390/s23010424_

Round 1
Reviewer 1 Report
The authors state in the caption of Figure 2: "Box plot Figure. Each left panel is the specifict index measured with baropodometric forceplate, instead, the right ones are those releaved with the SuPerSense Surface system. Green colour is the patient group (PG), violet is for elderly group (EG) and blu colour is for young group (YG). For each group, darkest colours are associated to open eyes condition, while light colours are for the closed eyes". But with the placement of dark and light colors it is difficult to understand. As the X axis already identifies the groups, I suggest placing only dark and light bars to characterize the subgroups.
Author Response
Reviewer #1
The authors state in the caption of Figure 2: "Box plot Figure. Each left panel is the specifict index measured with baropodometric forceplate, instead, the right ones are those releaved with the SuPerSense Surface system. Green colour is the patient group (PG), violet is for elderly group (EG) and blu colour is for young group (YG). For each group, darkest colours are associated to open eyes condition, while light colours are for the closed eyes". But with the placement of dark and light colors it is difficult to understand. As the X axis already identifies the groups, I suggest placing only dark and light bars to characterize the subgroups.
We thank the reviever1 for his/her overall positive comment on our manuscript.
According to all the reviewers’ suggestions, Figure 2 has been modified. In the right panel of CoP Ellipse Area, we deleted outlier points. We reworded the Figure 2 legend to allow a clear view of subgroups. Considering that boxplots are a bit squashed (especially for CoP Ellipse Area), it is not always easy to catch from color the specific condition. For this reason, we think it is better to keep both color and x-axis identifications for conditions.
Reviewer 2 Report
1. The opening line in the abstract should be the importance of the study.
2. There is a little discussion of the previous study using this device. Authors are requested to present in bullet points the contribution of this paper. It would be nicer to present how this study advances the field of postural assessment.
3. Was there any specific App used to collect the data? what types of sensor were used? are those weight sensors measure the capacitive or strain gauge? Pls add more details of the device, hardware and software control.
4. May be it would be easier to understand the SuperSense if a close view photo is presented. Also a 3D view or sketch to show the structure of the device and include some description of the working principle. Please describe the baropodometric methods/platform used to compare the study.
5. How the time was synchronized like how each OE and CE was recorded exactly for 51.2 seconds? Or was it collected for a longer time and than cropped? Was the stroke patients able to hold their eyes stable during the recording?
6. Can you pls add a figure or table for the raw data? and the processing algorithm/flow chart of the matlab script?
7. For figure 2, the data are good with small distribution except few outliers. May be it is better to post process the data without those outliers.
8. Pls add a schematic drawing to describe Length of COP path and COP ellipse area or any other terms used. may be from raw data? or from the statistical method?
Author Response
We thank Reviewer #2 for his/her overall positive comment on our manuscript. Below is the detailed response to each comment.

Round 2
Reviewer 2 Report
Authors have addressed my questions and it is ready to accept.